

# Integrated bioinformatics analysis of As, Au, Cd, Pb and Cu heavy metal responsive marker genes through *Arabidopsis thaliana* GEO datasets

Chao Niu[1,2,3,*], Min Jiang[2,3,*], Na Li[2,3,4], Jianguo Cao[4], Meifang Hou[1], Di-an Ni[1] and Zhaoqing Chu[2,3]

[1] School of Ecological Technology and Engineering, Shanghai Institute of Technology, Shanghai, Shanghai, China
[2] Shanghai Key Laboratory of Plant Functional Genomics and Resources, Shanghai Chenshan Botanical Garden, Shanghai, Shanghai, China
[3] Shanghai Chenshan Plant Science Research Center, Chinese Academy of Sciences, Shanghai, Shanghai, China
[4] College of Life Sciences, Shanghai Normal University, Shanghai, Shanghai, China
[*] These authors contributed equally to this work.

Corresponding authors
Di-an Ni, dani@sit.edu.cn
Zhaoqing Chu, zqchu@sibs.ac.cn

## ABSTRACT

**Background**. Current environmental pollution factors, particularly the distribution and diffusion of heavy metals in soil and water, are a high risk to local environments and humans. Despite striking advances in methods to detect contaminants by a variety of chemical and physical solutions, these methods have inherent limitations such as small dimensions and very low coverage. Therefore, identifying novel contaminant biomarkers are urgently needed.

**Methods**. To better track heavy metal contaminations in soil and water, integrated bioinformatics analysis to identify biomarkers of relevant heavy metal, such as As, Cd, Pb and Cu, is a suitable method for long-term and large-scale surveys of such heavy metal pollutants. Subsequently, the accuracy and stability of the results screened were experimentally validated by quantitative PCR experiment.

**Results**. We obtained 168 differentially expressed genes (DEGs) which contained 59 up-regulated genes and 109 down-regulated genes through comparative bioinformatics analyses. Subsequently, the gene ontology (GO) and Kyoto Encyclopedia of Genes and Genomes (KEGG) pathway enrichments of these DEGs were performed, respectively. GO analyses found that these DEGs were mainly related to responses to chemicals, responses to stimulus, responses to stress, responses to abiotic stimulus, and so on. KEGG pathway analyses of DEGs were mainly involved in the protein degradation process and other biologic process, such as the phenylpropanoid biosynthesis pathways and nitrogen metabolism. Moreover, we also speculated that nine candidate core biomarker genes (namely, *NILR1*, *PGPS1*, *WRKY33*, *BCS1*, AR781, *CYP81D8*, *NR1*, *EAP1* and *MYB15*) might be tightly correlated with the response or transport of heavy metals. Finally, experimental results displayed that these genes had the same expression trend response to different stresses as mentioned above (Cd, Pb and Cu) and no mentioned above (Zn and Cr).

**Conclusion**. In general, the identified biomarker genes could help us understand the potential molecular mechanisms or signaling pathways responsive to heavy metal stress

in plants, and could be applied as marker genes to track heavy metal pollution in soil and water through detecting their expression in plants growing in those environments.

# INTRODUCTION

There are naturally low concentrations of heavy metals in soil, but human activities such as mining, agriculture, sewage treatment and the metal industry have caused a sharp growth in their concentration in the environment, resulting in toxic effects on animals and plants (*Socha et al., 2015*). Therefore, heavy metal pollution is considered a source of significant environmental damage and a cause of increasing concern by the public and researchers. Many heavy metals are essential micronutrients for normal plant growth in trace amounts, such as iron (Fe), manganese (Mn), molybdenum (Mo), copper (Cu), zinc (Zn) and so on. However, excessive amounts can be toxic to plants, subsequently, the rest of the food chain. In addition, the accumulation of other heavy metals, such as cadmium (Cd), chromium (Cr), mercury (Hg), plumbum (Pb), aluminium (Al) etc., in plants can induce damage and toxicity (*Peralta-Videa et al., 2009*). The heavy metal is absorbed by the plant from the soil solution and transported to above-ground edible parts. It afterwards enters the animal or human body through the food chain, becoming a health threat (*Khan et al., 2013*). The key to controlling and repairing heavy metal pollution is the rapid and accurate detection of ones in order to determine the nature and extent of pollution. Many physical and chemical methods have been applied to estimate the concentration of heavy metals. For example, the prompt gamma ray neutron activation analysis (PGNAA) detection system has a minimum detectable concentration widely used for the determination of heavy metals by using a $^{241}$Am-Be neutron source and BGO detector (*Hei et al., 2016*). However, these solutions are not convenient nor efficient (*Bounakhla et al., 2012*). In addition, although optical and electrochemical detection are the two most widely used inspection methods in different fields, the accuracy of the former is lower than that of the traditional laboratory method, and the instrument is costly. The latter has more obvious inferiority on complicated sample pretreatment which may cause secondary pollution, overlapping interference of various heavy metals detection. Hence, our research's intention is to improve the techniques using higher sensitivity, a wider range of application and stronger specificity.

Indeed, many studies have reported that special genes can respond or transport to one or more heavy metals. For instance, it has been reported that six genes from three gene clusters *czcCBA1*, *cadA2R* and *colRS,* were involved with cadmium resistance in *Pseudomonas putida* CD2 (*Hu & Zhao, 2007*). While metal-responsive transcription factor 1 (MTF1) over-expression cells showed significantly delayed apoptosis, MTF1 null cells were susceptible to apoptosis in the presence of $Zn^{2+}$ or $Cd^{2+}$, which was augmented after

exposure to $Cr^{6+}$ (*Majumder et al., 2003*). OsNramp5 that belongs to natural resistance-associated macrophage protein (Nramp) families of metal transporters was a transporter for Mn acquisition and was the major pathway for Cd entry into rice roots (*Sasaki et al., 2012*). Another study for rice response to Cd stress using quantitative phosphoproteome yielded 2,454 phosphosites, associated with 1,244 proteins involved with signaling, stress tolerance, the neutralization of reactive oxygen species (ROS) and transcription factors (TFs) (*Zhong et al., 2017*). Over-expression of *ThWRKY7* can improve plant tolerance in *Tamarix hispida* (*Yang et al., 2016*) and ZmWRKY4 plays a vital role in regulating maize antioxidant defense under Cd stress (*Hong et al., 2017*). In addition, rice ASR1 and ASR5 act as transcription factors in concert and complementarily to regulate Al responsive genes (*Arenhart et al., 2016*). Moreover, co-application of 24-epibrassinolide (EBL) and salicylic acid (SA) resulted in the improvement of root and shoot lengths and chlorophyll content, which can regulate anti-oxidative defense responses and gene expression in *Brassica juncea* L. seedlings under Pb stress (*Kohli et al., 2018*). Metallothioneins (MTs) that are cysteine-rich, of low molecular weight, metal-binding proteins can enhance tolerance to heavy metals through differential responses in sweet potato (*Kim et al., 2014*), which may be strongly associated with Cd uptake, but not related to Cd transport from root to leaf, nor Cd enrichment in shoots (*Li et al., 2018*). Despite striking advances in gene function research involved with response or transport heavy metals, identifying novel accurate biomarkers to detect environmental pollutants is urgently needed.

The development of microarrays and high-throughput sequencing technologies has provided an integrated bioinformatics analysis method which could overcome the disadvantages of single traditional studies in deciphering critical genetic or physiological alternations and discovering promising biomarkers in screening for DEGs (*Vogelstein et al., 2013*). This approach would make it possible to analyze the associated pathways and interaction networks of the identified DEGs which have applied to predict the adverse drug reactions (ADRs) with the Library of Integrated Network-based Cellular Signatures (LINCS) L1000 data (*Duan et al., 2014*; *Wang, Clark & Ma'ayan, 2016*; *Subramanian et al., 2017*). Therefore, this approach has been applied extensively to the diagnosis and treatment of human cancers. For example, researchers believed that GNAI1, NCAPH, MMP9, AURKA and EZH2I were identified as the key molecules in patients with serous ovarian cancer resistant to carboplatin using integrated bioinformatics analysis (*Zhan, Liu & Hua, 2018*). COL1A2 as a diagnostic biomarker was highly tissue specific and was expressed in human gastric cancer by integrating bioinformatics and meta-analysis (*Rong et al., 2018*). In addition, a recent study has reported that 20 candidates were identified from 658 potential dehalogenases for exploration of protein functional diversity by integrating bioinformatics with expression analysis and biochemical characterization (*Vanacek et al., 2018*).

The identification of novel biomarkers is currently an efficient approach for large-scale screening and diagnosis (*Liu et al., 2018*; *Zhan, Liu & Hua, 2018*). Therefore, we performed analysis of 11 original microarray data to identify DEG response to different heavy metals in the present study. Additionally, functional enrichment analysis was further proposed to analyze the main biological functions and protein–protein interaction (PPI) networks

that were constructed to screen the crucial genes in response or transport heavy metals. At last, the expression trend between experimental validation and the microarray mentioned above was surveyed. Overall, these results suggested that highlighted key genes and pathways might be used as a biomarker to detect and further reduce heavy metal pollutants.

# MATERIAL AND METHODS

## Retrieval of gene expression profile data

Microarray data about heavy metal stresses on gene expression (GSE49037, GSE31977, GSE46958, GSE55436, GSE94314, GSE19245, GSE90701, GSE22114, GSE13114, GSE104916 and GSE65333) were downloaded from the Gene Expression Omnibus (GEO) (https://www.ncbi.nlm.nih.gov/geo/). It is reported that the size of the sample determines the reliability of the DEGs in microarray analysis and recent comprehensive bioinformatics research (*Moradifard et al., 2018*). Therefore, in order to obtain more complete available data, we have only chosen five kinds of heavy metal experiments which contained at least six samples, including arsenic (As), aurum (Au), Cd, Cu and Pb. The specific information of these experiment methods was explained in Table S1. For example, 8-days-old *A. thaliana* Col-0 plants were treated in 10 µM Cu and then harvested for further analyses after for treatment 24 h (Table S1). There are a total of 139 samples in present study, including 65 control samples and 74 heavy metal treatment samples (Table S1). Platform and series matrix file(s) were downloaded as TXT files. The dataset detailed information is shown in Table S1. The R software package (*R Core Team, 2018*) was used to process the downloaded files and to convert and reject the unqualified data. The data was calibrated, standardized, and $\log_2$ transformed.

## Identification of DEGs

The downloaded platform and series of matrix file(s) were converted using the R software package. The ID corresponding to the probe name was converted into the standard name as a gene symbol and saved in a TXT file. The limma package (*Ritchie et al., 2015*) in the bio-conductor package (http://www.bioconductor.org/) was performed to screen DEGs. The related operating instruction codes were put into R, and the DEGs in heavy metal treatment and control samples of every GEO datasets were analyzed. Genes with a corrected $P$-value $< 0.05$ and $|\log_2$fold change (FC)$| > 1$ were considered DEGs. The results were preserved as TXT files for subsequent analysis.

## Integration of data from different experiments

The list of DEGs from all series of heavy metal experiments was obtained by different expression gene analysis. Gene integration for the DEGs identified from the eleven datasets was executed using a robust rank aggregation (RRA) software package (*Kolde et al., 2012*). Based on the null hypothesis of irrelevant inputs, the RRA method filters out genes that are better aligned than expected. A list of genes obtained from the RRA approach that were up-or down-regulated in 20 raw data which were downloaded from GEO datasets as unqualified data that was usually used as source for Quantile normalization in R for subsequent analysis. The RRA algorithm detects genes that are ranked consistently better
than expected under null hypothesis of uncorrelated inputs and assigns a significance score for each gene. The RRA method is based on the hypothesis that each gene is randomly ordered in each experiment. If a gene ranked high in all experiments, then the smaller its *P*-value is, the greater the likelihood of differential gene expression. The RRA approach is openly available in the comprehensive R Network (http://cran.r.project.org/).

## Functional enrichment analyses of DEGs

In order to expound the underlying biological processes, molecular functions and cellular components connected with the overlapping DEGs identified above, GO enrichment analysis was executed using the GOATOOLS database for annotation (https://github.com/tanghaibao/goatools) (*Klopfenstein et al., 2018*). GOATOOLS was performed on the integrated DEGs using Fisher's accurate test. To ensure the minimum false positive rate, multiplex test methods such as Holm, Sidak and false discovery rate were used to correct the *P*-value. Once the corrected *P*-value < 0.05, the GO function was considered to have significant enrichment. Moreover, the KEGG pathway enrichment analysis of the overlapping DEGs was performed using the KOBAS online analysis database (http://kobas.cbi.pku.edu.cn) (*Xie et al., 2011*). The KOBAS analysis was based on the hypergeometric test/Fisher's exact test. Likewise, the corrected *P*-value < 0.05 was regarded as a statistically significant difference.

## PPI network integration based on STRING database

In order to better explore the physical contacts among protein molecules, the potential interactions among the overlapping DEGs were performed to identify the PPI network using the STRING database (http://string.db.org/) (*Szklarczyk et al., 2017*). PPIs were further imported to Cytoscape software for constructing the PPI network of overlapping DEGs (*Shannon et al., 2003*). Each node is a gene, protein, or molecule, and the connections between nodes represent the interaction of these biological molecules, which can be used for identifying interactions and pathway relationships between the DEGs about heavy metals. The corresponding proteins in the central node may be core proteins or key candidate genes which likely have vital physiological functions.

## Experimental validation

To better assess the veracity for the results identified above, we performed quantitative expression analysis of 20 candidate genes under different heavy metal stresses mentioned above (Cd, Pb and Cu) and not mentioned above (Zn and Cr) which can better explain the results' reliability. The real time quantitative PCR (RT-qPCR) experiments on RNA were collected from different time points (6 h, 12 h and 24 h) of 2 week-old *A. thaliana* Col-0 plants under 100 $\mu$M ZnSO4, 100 $\mu$M PbCl$_2$, 100 $\mu$M CrCl$_3$, 100 $\mu$M CuSO4 and 100 $\mu$M CdCl$_2$ treatments, respectively. The primer-sets were listed in Table S2.

# RESULTS

## GEO data information and identification of DEGs

The detailed information for the eleven GEO datasets contained in the current study is shown (Table 1, Table S1). In order to integrate different experiments and different

**Table 1  Detailed information about GEO data were showed in this study.**

| Dataset | Heavy metals | Platform | Number of samples (Treatment/ Control) | Ecotype | Tissue |
|---|---|---|---|---|---|
| GSE49037 | As | GPL198, GPL13970 | 15(9/6) | Col-0 | Roots |
| GSE31977 | As | GPL198 | 15(9/6) | Col-0, Ws-2 | Roots |
| GSE46958 | Au | GPL198 | 6(3/3) | Col-0 | Roots |
| GSE55436 | Au | GPL17416, GPL18349 | 12(6/6) | Col-0 | Roots |
| GSE94314 | Cd | GPL198 | 8(4/4) | Col-0, Bur-0 | Roots |
| GSE19245 | Cd | GPL198 | 24(12/12) | Col-0 | Roots, Shoots |
| GSE90701 | Cd | GPL14727 | 8(4/4) | Col-0 | Seedings |
| GSE22114 | Cd | GPL198 | 6(3/3) | Col-0 | Roots |
| GSE13114 | Cu | GPL1775 | 12(6/6) | Col-0 | Seedings |
| GSE104916 | Cu | GPL13222 | 21(12/9) | Col-0 | Roots, Rosettes |
| GSE65333 | Pb | GPL12621 | 12(6/6) | Col-0 | Roots, Shoots |

sequencing platforms, the data need to be processed and standardized. Therefore, the eleven GEO datasets were standardized (Fig. 1, Fig. S1). Then, these datasets were screened by the limma package with the condition, including the corrected $P$-value $< 0.05$ and $|\log_2 FC| > 1$. The results showed that there are 1,916, 2,555 and 1,147 DEGs identified from different As content treatments of GSE31977, respectively (Fig. 2A). While 2,030 and 776 DEGs were obtained from GSE49037 which contains As treatment with different content as well, respectively (Fig. S2). In addition, we screened 3,166 DEGs in GSE46958 with Au treatments (Fig. 2B), while 592 and 726 DEGs were obtained from different Au content treatments in GSE55436 (Fig. S2). As for the Cd treatments, we also obtained 1,832 and 972 DEGs in GSE94314; 486, 176, 410 and 377 DEGs in GSE19245; 124 and 154 DEGs in GSE90701, and 158 DEGs in GSE22114, respectively (Fig. 2C, Fig. S2). For Cu treatments, there were 110 and 91 DEGs in GSE13114; and 691, 495 and 485 DEGs in GSE104916 were obtained, respectively (Fig. 2D, Fig. S2). We also acquired 3,453 and 23 DEGs from GSE65333 with different Pb content treatments (Fig. 2E). Then, the cluster heat-map figure of the top 200 DEGs identified from each experiment was displayed by Multi-Experiment Viewer (MEV), respectively (Fig. 3, Fig. S3). Many gene expression levels showed significant difference in each treatment verse control (Fig. 3, Fig. S3).

## Identification of DEGs using integrated bioinformatics

The DEGs mentioned above have been screened by the limma package, and then were analyzed and extracted by RRA method according to the standard with the corrected $P$-value $< 0.05$ and $|\log_2 FC| > 1$. It is noteworthy that the RRA method is based on the hypothesis that each gene is randomly ordered in each experiment. That is, if a gene ranked high in all experiments, then the smaller its $P$-value was, and the greater the likelihood of differential gene expression. Our rank analysis results showed that a total of 168 DEGs comprising 109 down-regulated and 59 up-regulated

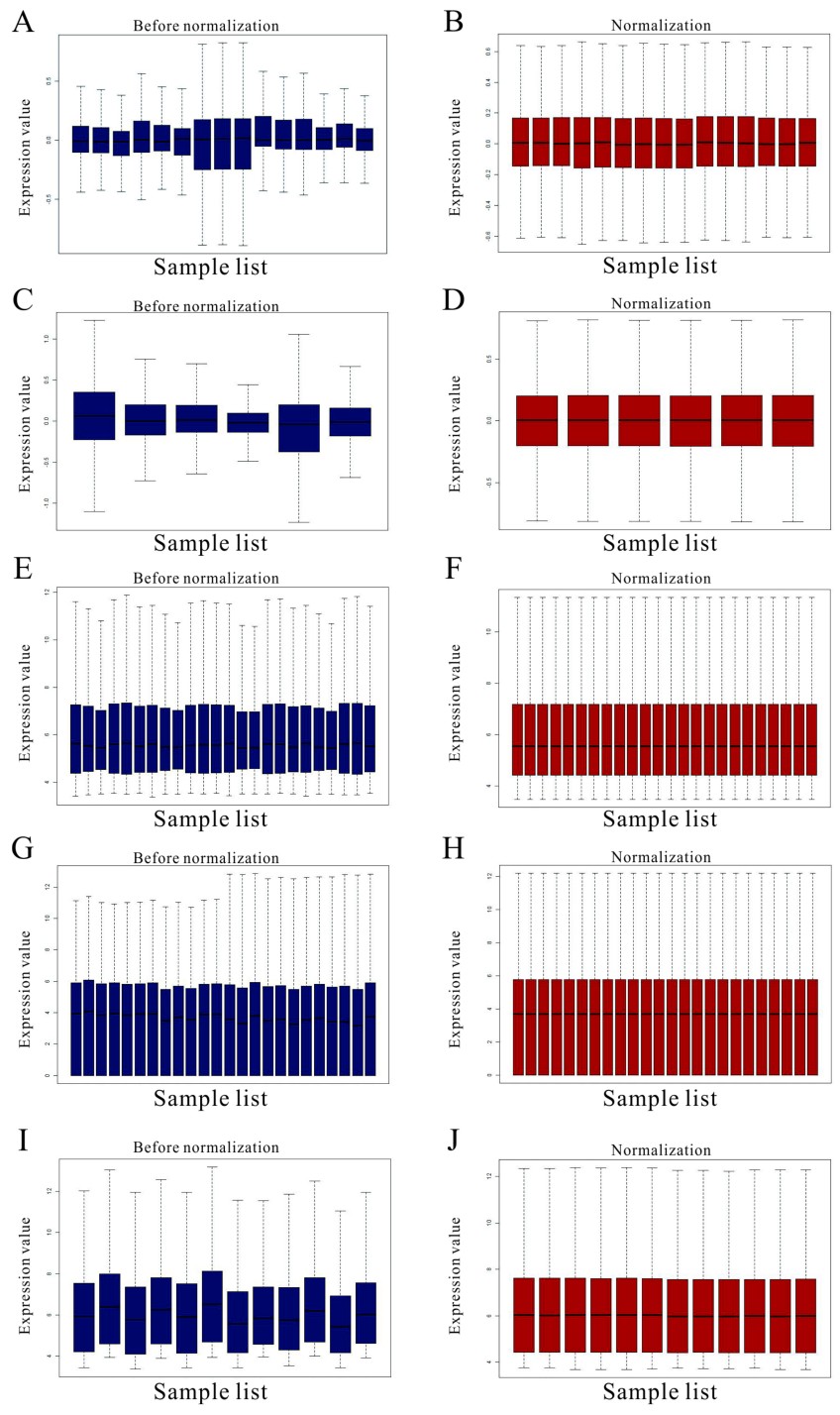

**Figure 1  Standardization for all samples included GEO datasets.** (A–B) The standardization of GSE31977 data; (C–D) The standardization of GSE46958 data; (E–F) The standardization of GSE19245 data; (G–H) The standardization of GSE104916 data; (I–J) The standardization of GSE65333 data. The blue bar represents the data before normalization, and the red bar represents the normalized data.

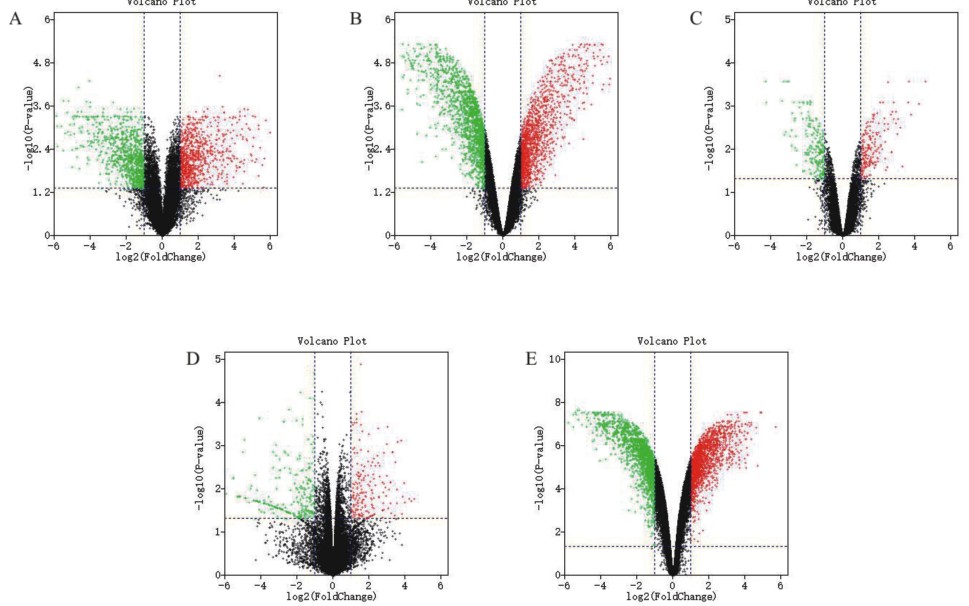

**Figure 2** **Differential expression analysis of datasets of samples.** (A) GSE31977 data; (B) GSE46958 data; (C) GSE19245 data; (D) GSE104916 data; (E) GSE65333 data. The red points represent up-regulated genes screened on the basis of |fold change | > 2.0, $P$-value < 0.05. The green points represent down-regulated genes screened on the basis of |fold change | > 2.0, $P$-value < 0.05. The black points represent genes with no significant difference. FC is the fold change.

genes were obtained (Table 2, Tables S3 and S4). The top 20 most significantly up-regulated genes were *AT3G46270, ATCSLB05, AT3G19030, COL9, BCAT4, ELF4, CYP83A1, AT1G76800, AT1G61740, CLE6, AT4G01440, AT1G72200, MOT1, AT5G52790, AT4G40070, AT4G25250, EXGE-A1, NR1, AT5G19970* and *CYP735A2* (Table 2). Additionally, the top 20 most significantly down-regulated genes were *DIN2, WRKY75, AT4G15120, CYP81F2, AT1G73480, AT1G72900, PGPS1, AT5G06730, AT1G35910, CYP81D8, AT3G12320, ATERF6, AT1G12200, AT5G25450, AT4G28460, NILR1, HSP70, APRR9, Fes1A* and *AT3G02800* (Table 2). Moreover, the heatmap figure of the top 20 up- and down-regulated genes was performed using R-heatmap software (Fig. 4). For instance, *PGPS1* and WRKY75 were all down-regulated in five treatments, while *CYP83A1* and *NR1* showed up expression trend in several conditions (Fig. 4).

## GO term enrichment analysis of DEGs

To illuminate the potential biological functions of DEGs, GO annotation of the integrated DEGs mentioned above was executed by the GOATOOLS and GO functional enrichments of up- and down-regulated genes with the standard of corrected $P$-value < 0.05 (Table S5). These results exhibited that they were significantly enriched in multiple biological processes: physiological process (GO: 0008150), cellular physiological process (GO: 0009987), physiological response to stimulus (GO: 0050896), response to biotic/abiotic stress (GO: 0006950) and so on (Fig. 5A). Meanwhile, the cellular component exhibited a close correlation with the cellular (GO: 0005575) and extracellular components (GO:
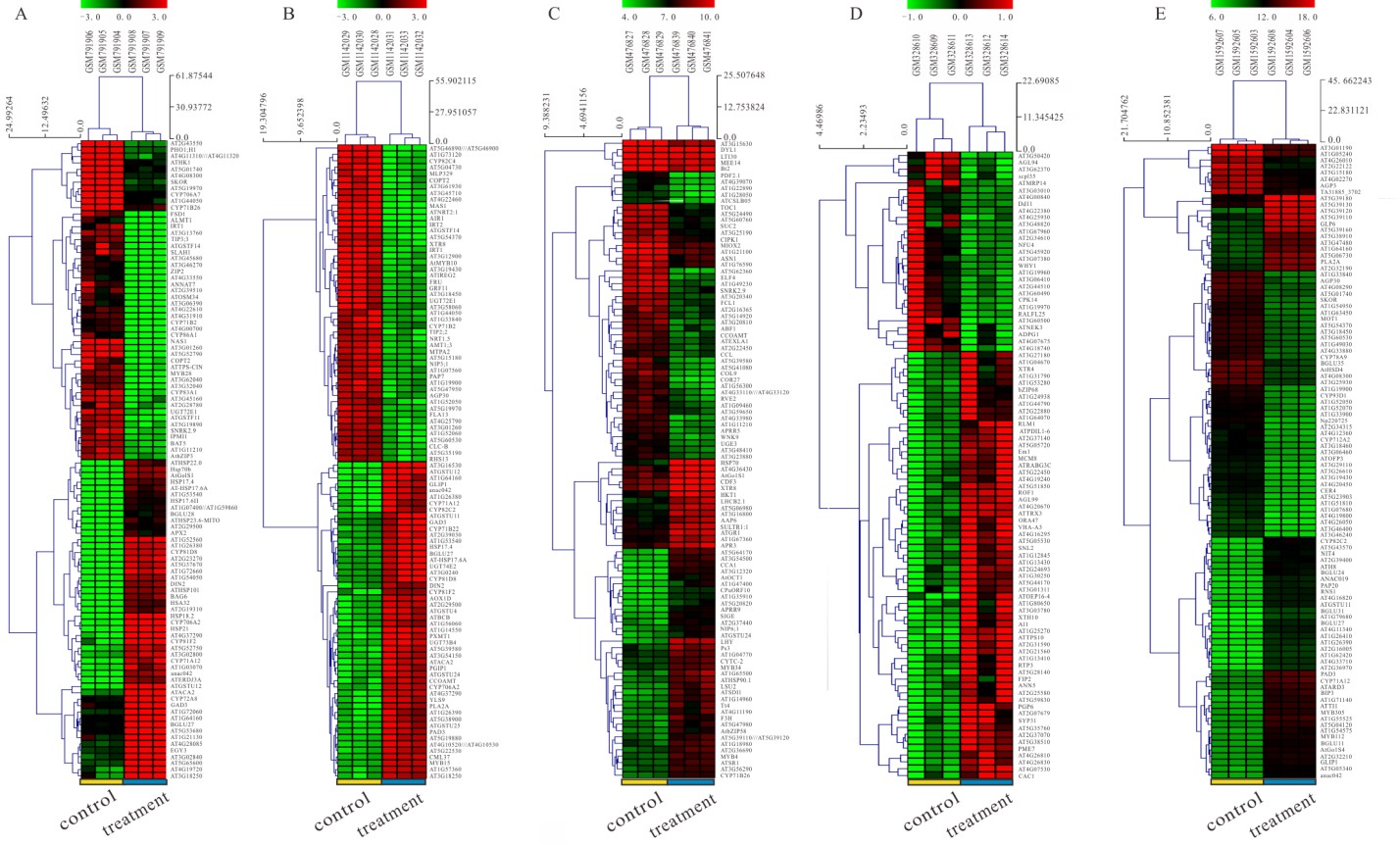

**Figure 3** Hierarchial clustering heatmap image of DEGs screened on the basis of |fold change | > 2.0, *P*-value < 0.05. (A) GSE31977 data (As treatment), (B) GSE46958 data (Au treatment), (C) GSE19245 data (Cd treatment), (D) GSE13114 data (Cu treatment), (E) GSE65333 data (Pb treatment).

0005576) (Fig. 5A). As for the 59 up-regulated genes, they showed a close correlation with membrane, such as the membrane part (GO: 0044425), intrinsic component of membrane (GO: 0031224), membrane (GO: 0016020), integral component of membrane (GO: 0016021) and so on (Table 3). In terms of the 109 down-regulated genes, they were significantly enriched in multiple biological processes related to environment stress; for example, in response to stimulus (GO: 0050896), in response to stress (GO: 0006950), in response to chemicals (GO: 0042221) and so on (Table 3). Moreover, the GO term enrichments of integrated DEGs were mainly enriched in response to chemicals, abiotic stimulus, oxygen-containing compounds, organic substances, external stimulus and so on, all of which are involved with plant functions in response to stress (Fig. 5B).

## KEGG pathway analysis of DEGs

To better understand the function enrichment of the identified DEGs, the KOBAS online analysis database was used. The results exhibited that the most significantly enriched pathways of the DEGs are mainly involved with protein degradation, such as bisphenol degradation, polycyclic aromatic hydrocarbon degradation, limonene and pinene

**Table 2  Screening DEGs in *A. thaliana* treated with heavy metal by integrated GEO data.**

| DEGs | Gene names |
| --- | --- |
| Up-regulated | AT3G46270 ATCSLB05 AT3G19030 COL9 BCAT4 ELF4 CYP83A1 AT1G76800 AT1G61740 CLE6 AT4G01440 AT1G72200 MOT1 AT5G52790 AT4G40070 AT4G25250 EXGT-A1 NR1 AT5G19970 CYP735A2 AT3G59370 AT3G25790 PIP2;4 AT4G16447 AT1G22500 NIR1 AT5G06610 CER4 AT1G12080 AT4G31910 LAX3 AT1G33340 LCR69 UCC1 ENODL8 GA3OX1 AT4G39070 CYP93D1 AT1G21440 AT4G01140 CKX4 AT3G12900 RVE2 AT3G20015 AT1G09750 AT2G40900 AT3G23880 PIP1;5 AGP4 AT3G32040 AT2G01900 AT1G51820 AIR1 AT5G04730 AT5G42860 BT2 CYP71B2 AT5G26280 AT1G24800 AT1G24881 AT1G25055 AT1G25150 AT1G25211 |
| Down-regulated | DIN2 WRKY75 AT4G15120 CYP81F2 AT1G73480 AT1G72900 PGPS1 AT5G06730 AT1G35910 CYP81D8 AT3G12320 ATERF6 AT1G12200 AT5G25450 AT4G28460 NILR1 HSP70 APRR9 Fes1A AT3G02800 AT2G23270 CRK11 AT3G07090 AT5G26220 AT5G05220 ATSDI1 ATGSTU4 AT1G61340 ERF2 AT5G64510 ATHSP23.6-MITO BCS1 MBF1C ATGSTF6 AT3G62550 AT3G09440 ZIFL1 AT1G12030 AT5G05340 PXMT1 AT5G02230 AT5G02490 ACS6 GLIP1 AT4G16260 AT5G13200 AT2G18670 AT2G18680 AT5G39110 AT5G39120 AT5G39150 AT5G39180 AT5G20820 AT5G64170 AT4G28350 AT3G47540 AT2G19310 CSLE1 CYP710A1 WRKY45 AT3G59080 CYP706A2 AT4G19810 AT4G04330 AT2G21640 ATHSFA2 AT1G72940 AT2G35730 AT4G37290 MYB51 AT1G60750 AT5G20910 AT1G72060 AR781 ATBCB ABA1 ORG1 YLS9 AT4G15420 AT3G09405 AT5G06760 AT5G53970 ICL AT5G51440 BCA3 SBT3.5 AT4G39830 MYB15 AT5G52750 AAC3 AT3G48510 HSP17.4 AT5G42830 AT5G48000 AT1G71140 AtPP2-A11 WRKY33 SULTR1;1 AT3G50910 AT2G36460 AT2G48090 CEJ1 CHI AT1G14550 PAP20 AT5G14730 NAPRT2 CRK10 GRX480 AT5G39580 DMP1 AT3G12510 AT3G48450 |

degradation, aminobenzoate degradation and so on (Table S6 and Fig. 6). Moreover, KEGG pathway analysis demonstrated that the DEGs were significantly enriched in the estrogen signaling pathway, zeatin biosynthesis, measles, antigen processing and presentation, MAPK signaling pathway and nitrogen metabolism (Table S6 and Fig. 6).

## PPI network and modules analysis

In order to further investigate the interactions and networks of these DEGs, the PPI network was identified using the STRING database and constructed using Cytoscape software, which consisted of 111 nodes and 402 interactions (Fig. 7, Tables S7 and S8). Identification of candidate hub nodes generally are needed to calculate the topological features, including degree (*Williams & Del Genio, 2014*) and betweenness (*Agryzkov et al., 2014*). The importance degree of a gene is proportional to the size of the two quantitative

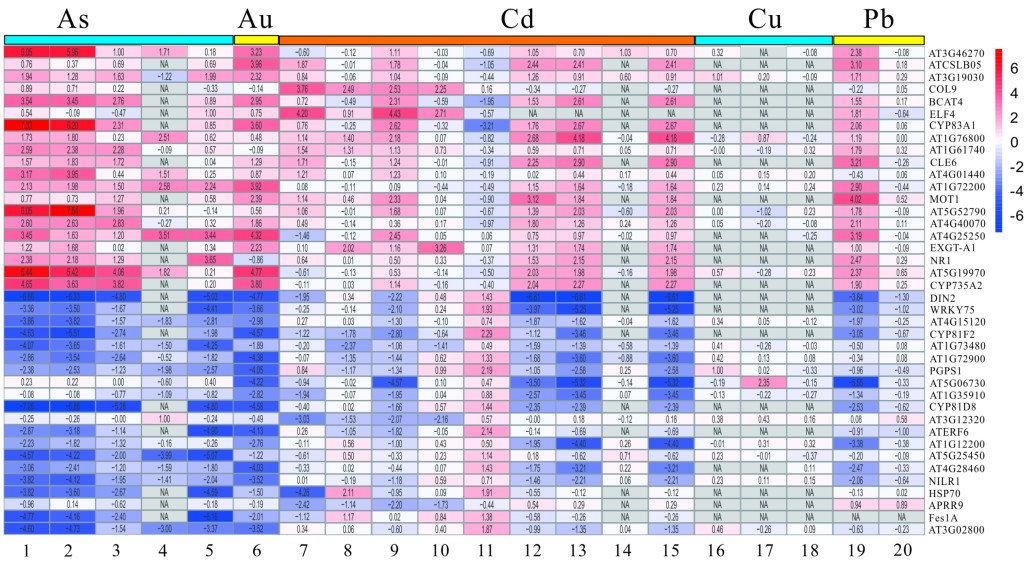

**Figure 4** **The LogFC heatmap image of each kind of treatment.** The abscissa is the GEO ID and Lowercase letters represent one kind of content heavy metal treatment included, and the ordinate is the gene name. The value in the box is the logFC value, and "NA" represent the *P*-value >= 0.05.

values in this network. Therefore, 9 candidate hub nodes were preliminary selected, namely nematode-induced LRR-RLK 1 (*NILR1*), 3-phosphatidyl-1′-glycerol-3′-phosphate synthase 1 (*PGPS1*), WRKY DNA-binding protein 33 (*WRKY33*), cytochrome bc1 synthase 1 (*BCS1*), pheromone receptor-like protein (*AR781*), "cytochrome P450, family 81, subfamily D, polypeptide 8" (*CYP81D8*), nitrate reductase 1 (*NR1*), eukaryotic aspartyl protease 1 (*EAP1*) and MYB domain protein 15 (*MYB15*) (Fig. 7 and Table S8).

## Expression analysis of screened key genes under Cd, Pb, Cu, Zn and Cr heavy metal stresses

In order to better calculate the reliability of the preliminary selected results, the RT-qPCR analysis of 20 candidate genes under different heavy metal stresses containing those mentioned above (Cd, Pb and Cu) and not mentioned above (Zn and Cr) was performed. As expected, these genes showed up- or down-regulated trends in one or several heavy metal treatments (Fig. 8). Specially, most genes have obvious high expression levels under Cu condition, while moderate down-regulated under Pb and Cr treatments (Fig. 8). Moreover, several genes exhibited distinct down-regulated under most situation, for example, HSP70, AT5G52750, BCS1 and MYB15 (Fig. 8). Our results indicated that these genes selected above should respond to heavy metal stress in plants and it is possible to use them as biomarker.

## DISCUSSION

### Uptake and analyses of DEGs from GEO datasets

Microarray analysis of gene expression profiles is widely used in distinguishing disease-related genes and biological pathways. However, such studies on heavy metal stresses

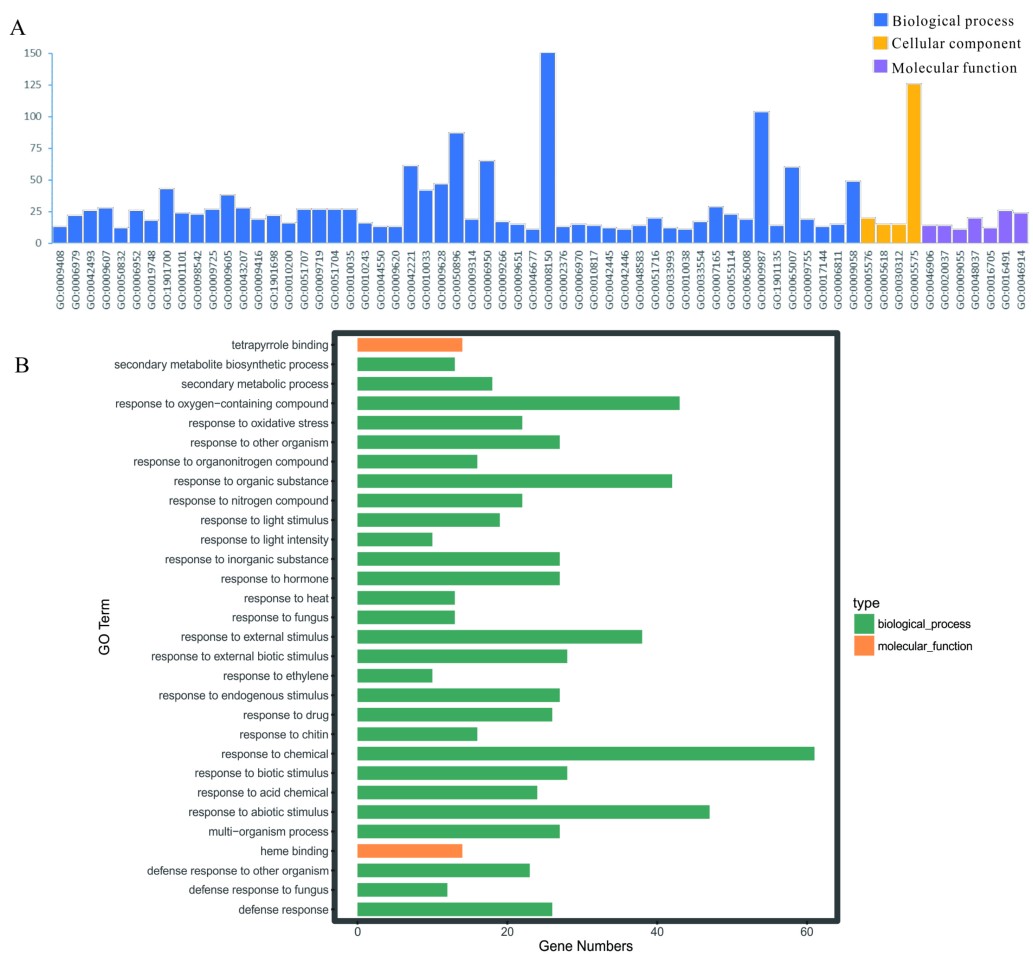

**Figure 5** **GO enrichment analysis of the screened DEGs.** (A) GO enrichment divided DEGs into three functional group: molecular function, biological processes, and cell composition. (B) GO enrichment significance items of DEGs in different functional groups.

have been scarce to date. Thus, in the present study, we identified 168 overlapping DEGs, including 109 down-regulated and 59 up-regulated genes, in the eleven microarray datasets involving heavy metal stresses (Table 2). Moreover, the GO analysis results suggested that the overlapping genes were mostly involved in the physiological process, cellular physiological process, response to biotic/abiotic stress and physiological response to stimulus at the levels of biological processes (Fig. 5A). For the up-regulated genes, their functions are mainly associated with the membrane, including the membrane part, intrinsic component of membrane, membrane, integral component of membrane and so on (Table 3). In terms of the down-regulated genes, they mainly play important roles in various responses to environmental stresses, including response to stimulus, response to stress, and response to chemicals (Table 3). Furthermore, KEGG pathway enrichment analysis showed that the overlapping DEGs were enriched significantly within protein degradation, including bisphenol degradation, polycyclic aromatic hydrocarbon degradation, limonene

**Table 3  GO analysis of DEGs associated with *A. thaliana* treated by heavy metal.**

| DEGs | Term | Description | Count | *P*-value |
|------|------|-------------|-------|-----------|
| Up-regulated | GO:0044425 | membrane part | 23 | 1.58E-05 |
| | GO:0031224 | intrinsic component of membrane | 22 | 9.67E-06 |
| | GO:0050896 | response to stimulus | 22 | 0.00316 |
| | GO:0016020 | membrane | 19 | 0.0367 |
| | GO:0016021 | integral component of membrane | 17 | 0.00135 |
| | GO:0046872 | metal ion binding | 17 | 0.00487 |
| | GO:0043169 | cation binding | 17 | 0.00499 |
| | GO:0042221 | response to chemical | 15 | 0.000519 |
| | GO:0046914 | transition metal ion binding | 14 | 9.27E-05 |
| | GO:0016491 | oxidoreductase activity | 12 | 0.000621 |
| | GO:1901700 | response to oxygen-containing compound | 11 | 0.000636 |
| | GO:0010033 | response to organic substance | 11 | 0.00147 |
| | GO:0009628 | response to abiotic stimulus | 11 | 0.0153 |
| | GO:0009725 | response to hormone | 10 | 0.000345 |
| | GO:0009719 | response to endogenous stimulus | 10 | 0.000398 |
| | GO:0065008 | regulation of biological quality | 10 | 0.000906 |
| | GO:0009605 | response to external stimulus | 9 | 0.00866 |
| Down-regulated | GO:0009987 | cellular process | 69 | 0.00696 |
| | GO:0008152 | metabolic process | 68 | 0.00512 |
| | GO:0050896 | response to stimulus | 67 | 1.03E-06 |
| | GO:0044237 | cellular metabolic process | 55 | 0.024 |
| | GO:0006950 | response to stress | 54 | 7.77E-07 |
| | GO:0042221 | response to chemical | 48 | 6.03E-07 |
| | GO:0043231 | intracellular membrane-bounded organelle | 45 | 0.0385 |
| | GO:0065007 | biological regulation | 38 | 0.0279 |
| | GO:1901363 | heterocyclic compound binding | 38 | 0.0493 |
| | GO:0097159 | organic cyclic compound binding | 38 | 0.0495 |
| | GO:0009628 | response to abiotic stimulus | 37 | 1.04E-06 |
| | GO:0050789 | regulation of biological process | 35 | 0.0207 |
| | GO:1901700 | response to oxygen-containing compound | 34 | 3.87E-07 |
| | GO:0050794 | regulation of cellular process | 32 | 0.0234 |
| | GO:0009058 | biosynthetic process | 32 | 0.0316 |
| | GO:0010033 | response to organic substance | 31 | 5.54E-07 |
| | GO:0009605 | response to external stimulus | 30 | 5.16E-07 |
| | GO:1901576 | organic substance biosynthetic process | 30 | 0.0398 |
| | GO:0009607 | response to biotic stimulus | 24 | 3.55E-07 |
| | GO:0043207 | response to external biotic stimulus | 24 | 9.29E-07 |

and pinene degradation, and aminobenzoate degradation, which indicated these gene might be important in detoxication and transport of heavy metals (Table S6 and Fig. 6). Additionally, the estrogen signaling pathway, zeatin biosynthesis, MAPK signaling pathway, nitrogen metabolism, phenylpropanoid biosynthesis were analyzed as the major pathways of the important modules of overlapping DEGs. A recent study has shown that proteins

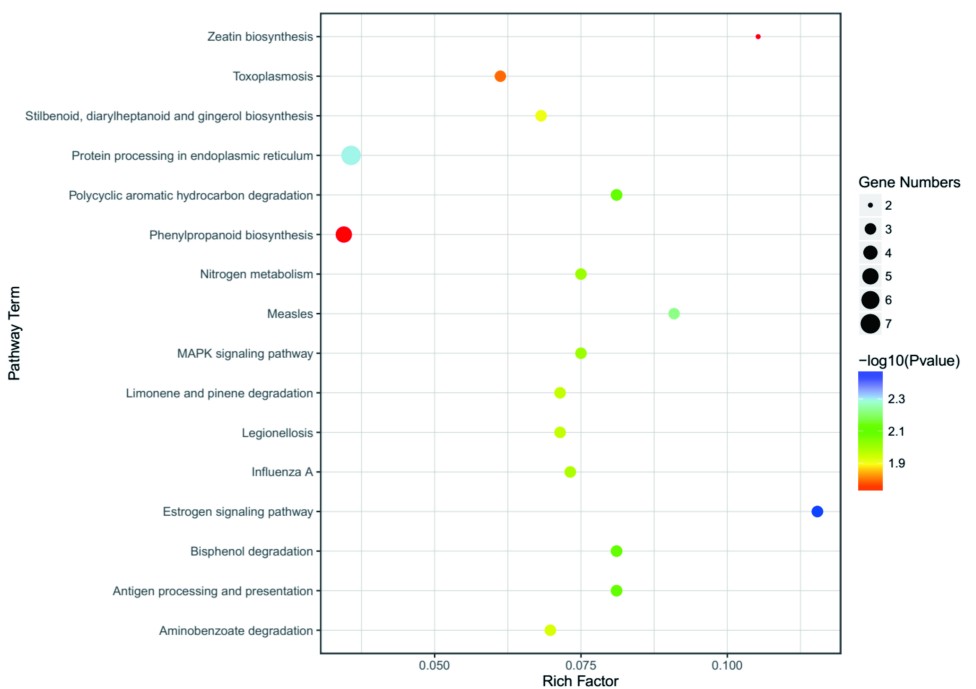

**Figure 6    KEGG enrichment analysis of the screened DEGs.** The abscissa is the Rich Factor, and the ordinate is the KEGG pathways ID. Circle size represents the number of genes, and circle color represents the *P*-value.

associated with oxygen metabolism, phenylpropanoid biosynthesis, and redox reaction were resistant to stresses such as high temperature, chilling, light and wounding (*Vogt, 2010*). Likewise, many plant *MKK* genes are induced by abiotic stresses, including wounding, drought, genotoxicity, cold, osmosis, high salinity, heat, and oxidative and UV-B treatment (*Jiang & Chu, 2018*). Another study has shown that plants can respond to heavy metal stress by induction of several different MAPK pathways, and that excessive copper and cadmium ions lead to distinct cellular signaling mechanisms in roots (*Jonak, Nakagami & Hirt, 2004*). These enriched pathways provide insights into the molecular mechanism of heavy metal uptake, transport and metabolism. Therefore, it can help in the development of new biological detection strategies.

## Identification of candidate core genes under heavy metal stresses

We also identified nine major hub genes according to the PPI network and our experimental validation, namely *NILR1*, *PGPS1*, *WRKY33*, *BCS1*, AR781, *CYP81D8*, *NR1*, *EAP1* and *MYB15* (Figs. 7 and 8). As is all known, plant activators are chemicals that induce plant defense responses to a broad spectrum of pathogens. *NILR1*, a new potential plant activator (PPA) with high expression levels after PPA treatment, enhanced plant defense ability against pathogen invasion through the plant redox system (*Matsushima & Miyashita, 2012*). Moreover, *NILR1*, a Leu-rich repeat transmembrane receptor protein kinase (LRR-RLK), was found recently in human mitochondria (*Heazlewood et al., 2004*). Furthermore, over-expression of *PGPS1* can enhance tolerance to oxidative stress in

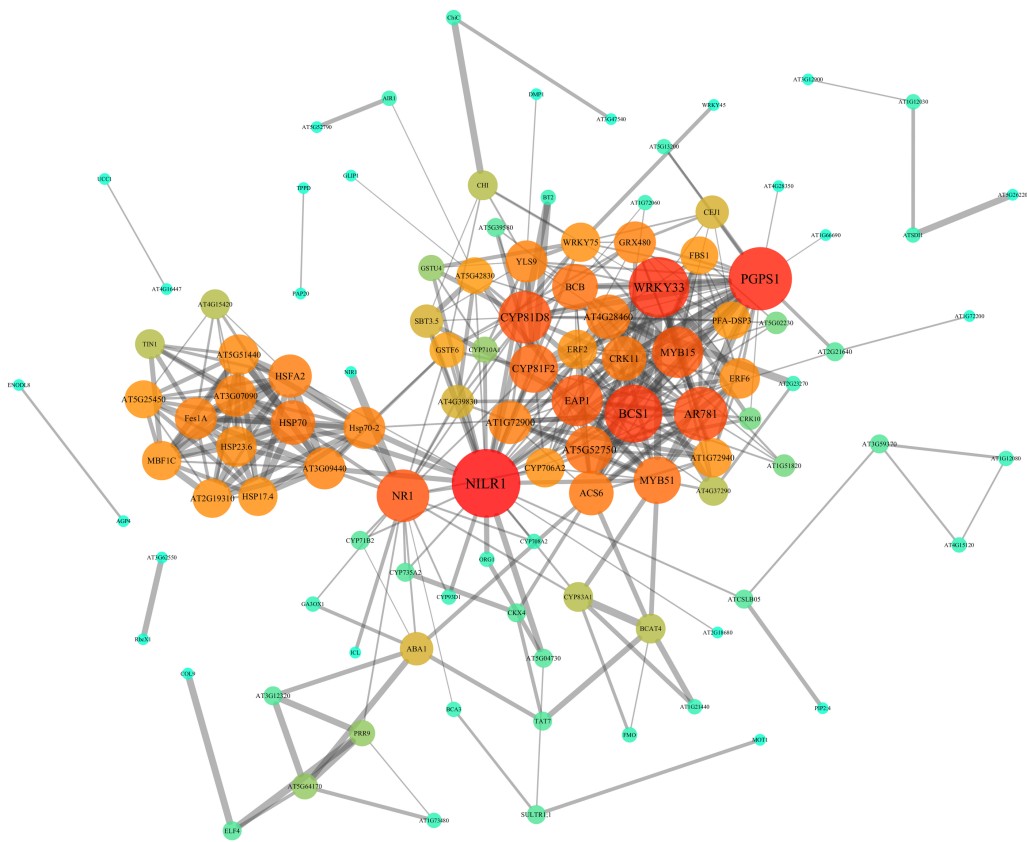

**Figure 7  PPI network of identified DEGs constructed by the STRING database.** Circles represent genes, lines represent the interaction of proteins between genes, and the results within the circle represent the structure of proteins. Line thickness and circle color represents the degree of significance between the proteins.

transgenic Arabidopsis plants (*Luhua et al., 2008*) and plays a role in early transcriptional defense responses and reactive oxygen species (ROS) production in Arabidopsis cell suspension culture under high-light conditions (*Gonzalez-Perez et al., 2011*). While ROS function as signal transduction molecules in the acclimation process of plants when exposed to abiotic stress factors, such as drought, heat, salinity and high light (*Choudhury et al., 2017*). It is known to all that LRR-RLK genes are often the first to perceive external environmental changes through general ROS production, $Ca^{2+}$ signature, etc., while the toxic effects of heavy metals usually cause ROS production. Hence, it could be understood that *NILR1* and *PGPS1* responds to environmental stress, especially heavy metal stresses.

Over-expression of WRKY33 can increase tolerance to NaCl in Arabidopsis, while increasing sensitivity to ABA (*Jiang & Deyholos, 2009*). In addition, WRKY33 negatively regulated ABA signaling in response to pathogenic bacteria (*Liu et al., 2015*). Furthermore, WRKY33 that is induced by pathogen infection, salicylate signaling or the paraquat herbicide that generates activated oxygen species in exposed cells, is an important transcription factor that plays a critical role in response to necrotrophic pathogens through

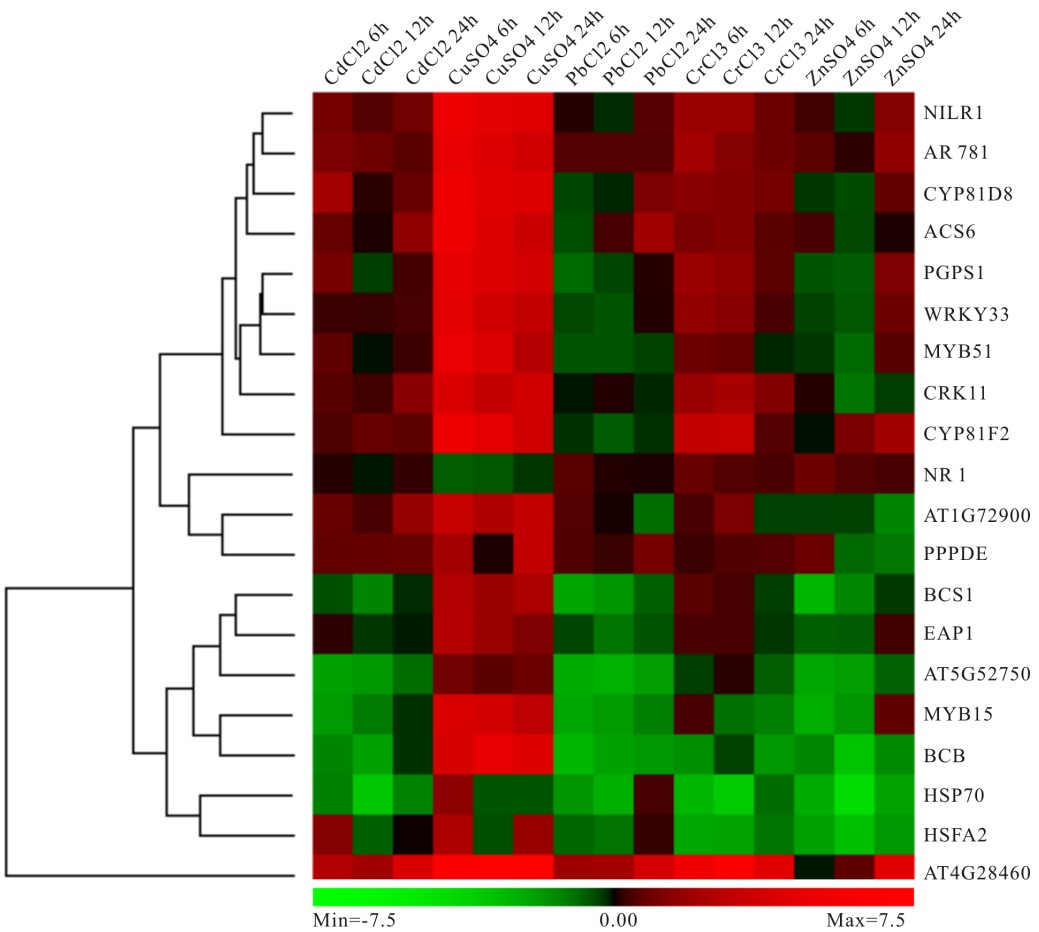

**Figure 8** Experimental validation of selected candidate key genes.

interaction with ATG18a (*Zheng et al., 2006*; *Lai et al., 2011*). BCS1 is re-annotated as ATPase OM66 (Outer Mitochondrial membrane protein of 66 kDa) because it harbors the outer mitochondrial membrane and lacks the BCS1 domain. AtOM66 over-expression in transgenetic plants leads to more tolerance to drought stress, accelerated cell death rates, increased SA content and more susceptibility to the necrotrophic fungus (*Zhang et al., 2014*). It is interesting that *AtOM66* and *AtWRKY33* are involved with the transcriptional response to MV (methyl viologen) which is triggered by chloroplast-generated superoxide signaling (*Van Aken et al., 2016*), and had an interaction in the PPI network (Fig. 7). Moreover, they are also involved in the transcriptional innate immune response to flg22 (*Navarro et al., 2004*). AR781 is involved in the early elicitor signaling events which occur within minutes and include ion fluxes across the plasma membrane, activation of MPKs and the formation of ROS related to *PGPS1* and *WRKY33* mentioned above (*Benschop et al., 2007*). CYP81D8 is located in the ER (*Dunkley et al., 2006*), which is consistent with the results of KEGG enriched analysis (Table S6 and Fig. 6). Additionally, CYP81D8, which are KAR1 (Karrikins) up-regulated genes, are enriched for light-responsive transcripts during germination and seedling development in *A. thaliana* (*Nelson et al., 2010*).

NR1 (also called NIA1) that mediates NO synthesis in plants are critical to abscisic acid (ABA)-induced stomatal closure of guard cells (*Desikan et al., 2002*). Specifically, NR1 and NR2 control stomatal closure by altering genes of core ABA signaling components to regulate $K^+$ in currents and ABA-enhanced slow anion currents in Arabidopsis (*Zhao et al., 2016*). Moreover, the activity of NR1 can be dramatically increased by being sumoylated by the E3 SUMO ligase AtSIZ1 (*Park, Song & Seo, 2011*), while its nitrate-responsive expression was controlled by its regulatory region, including the 3′-untranslated region, and 5′-and 3′-flanking sequences (*Konishi & Yanagisawa, 2011*). EAP1 that has aspartic-type endopeptidase activity evolved in response to their environments and ecosystems through an investigated proteome-wide binary protein-protein interaction network (*Braun et al., 2011*). MYB proteins are a superfamily of transcription factors that play important roles in many physiological processes and defense responses, such as salt stress, ethylene, auxin, jasmonic acid, chitin and so on (*Chen et al., 2006*). MYB15 is phosphorylated by MPK6, negatively regulating the expression of CBF (C-repeat-binding factor) genes which are required for freezing tolerance in Arabidopsis (*Kim et al., 2017*). MYB15, SIZ1, HOS1 (a RING-type ubiquitin E3 ligase) and ICE1 (MYC-like basic helix-loop-helix transcription factor) form a dynamic regulation network in response to freezing stress in Arabidopsis (*Miura et al., 2007*). In addition, over-expression of MYB15 confers drought and salt tolerance in Arabidopsis, which is possibly performed by enhancing the expression levels of the ABA biosynthesis and signaling related genes and those stress-protective proteins (*Ding et al., 2009*). Furthermore, MYB15 also controls defense-induced basal immunity and lignifications that are used in the conserved basal defense mechanism in the plant innate immune response (*Chezem et al., 2017*) and regulates stilbene biosynthesis involvement with MYB14 in grapevines whose key enzymes responsible for resveratrol biosynthesis are stilbene synthases (STSs) (*Holl et al., 2013*).

## Comparison with other existing related high-throughput methods

To date, a variety of high-throughput methods for identifying key signaling or pathways have been developed by investigators. They all can accurately detect only based on different models or systems. For example, drug-induced adverse events prediction combining chemical structure (CS) and gene expression (GE) features which is from the Library of Integrated Network-based Cellular Signatures (LINCS) L1000 dataset has been proposed and validated (*Wang, Clark & Ma'ayan, 2016*). Such as, WRKY, MYB, NR1 and NILR have been found in this database, which indicated that the results identified were reliable (Fig. 7). Quantitative structure activity relationship (QSAR) models together with the state-of-the-art machine learning techniques is faster and cheaper than large-scale virtual screening methods to identify the chemical compounds (*Soufan et al., 2018*). Another novel multi-label classification (MLC) technique using Bayesian active learning to mine large high-throughput screening assays has been proposed (*Soufan et al., 2016*). Moreover, a reduced transcriptome approach to monitor potential effects by environmental toxicants at genome-scale using zebrafish embryo test was also developed, whose reliability was assessed by RNA-ampliseq technology to identify DEGs (*Wang et al., 2018*). Furthermore, previous study have usually focused on the aspects of cancer diagnosis and treatment using

integrated bioinformatics analyses, with only a few other related research studies (*Cao et al., 2018*; *Zhan, Liu & Hua, 2018*), while we also propose a novel computational method to identify hub genes based on the same DEGs under different status at genome-scale. Subsequently, we verified the feasibility of our method through quantitative analysis of key gene expression (Fig. 8). As expected, almost all genes showed differently expression under one or several heavy metal stresses (Fig. 8). The analysis results indicated that our method is feasible for screening hub genes, and can be used as an effective supplementary tool for future ecological restoration research using traditional methods.

## CONCLUSION

A total of 168 overlapping DEGs were screened from 11 GEO datasets using integrated bioinformatics analysis approaches, including 59 up-regulated genes and 109 down-regulated genes. GO and KEGG pathway enrichment analysis revealed that DEGs were mainly enriched in nitrogen metabolism, polycyclic aromatic hydrocarbon degradation, antigen processing and presentation, MAPK signaling pathway and phenylpropanoid biosynthesis in plant responses to stress, which can provide a theoretical basis for studying the biological processes of heavy metals. Moreover, we successfully constructed a PPI network of DEGs underlying heavy metals and identified nine core genes (*NILR1*, *PGPS1*, *WRKY33*, *BCS1*, AR781, *CYP81D8*, *NR1*, *EAP1* and *MYB15*) which might be involved in the response to abiotic stress factors, particularly heavy metals and experimental validation were obtained the similar expression profiles under several heavy metal treatments, even no mentioned above (Cr and Zn). These findings should provide some clues for the future detection of heavy metal pollutants in water and soil. However, further molecular biological experimental studies are urgently required to validate the functions of candidate hub genes associated with heavy metals because our study was only performed based on data analysis.

### Funding
This work was funded by grants from the Shanghai Landscaping Administrative Bureau (Grant Nos. G182405 and G192412). The funders had no role in study design, data collection and analysis, decision to publish, or preparation of the manuscript.

### Grant Disclosures
The following grant information was disclosed by the authors:
Shanghai Landscaping Administrative Bureau: G182405, G192412.

### Competing Interests
The authors declare there are no competing interests.

### Author Contributions
- Chao Niu conceived and designed the experiments, performed the experiments, analyzed the data, prepared figures and/or tables, authored or reviewed drafts of the paper, approved the final draft.

- Min Jiang conceived and designed the experiments, performed the experiments, analyzed the data, contributed reagents/materials/analysis tools, prepared figures and/or tables, authored or reviewed drafts of the paper, approved the final draft.
- Na Li performed the experiments, analyzed the data, contributed reagents/materials/-analysis tools, approved the final draft.
- Jianguo Cao and Meifang Hou analyzed the data, contributed reagents/materials/analysis tools, approved the final draft.
- Di-an Ni analyzed the data, approved the final draft.
- Zhaoqing Chu conceived and designed the experiments, approved the final draft.

## Data Availability

We calculated and analyzed public GEO data using integrated bioinformatics analysis: GSE49037; GSE31977; GSE46958; GSE55436; GSE94314; GSE19245; GSE90701; GSE22114; GSE13114; GSE104916; GSE65333.

## Supplemental Information

Supplemental information for this article can be found online at http://dx.doi.org/10.7717/peerj.6495#supplemental-information.

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
