# Peer review of "Integrated bioinformatics analysis of As, Au, Cd, Pb and Cu heavy metal responsive marker genes through Arabidopsis thaliana GEO datasets"

_PeerJ, doi:10.7717/peerj.6495_

## Round 0.1 · original submission · Major Revisions

Although one of the reviewers has recommended that your manuscript be rejected, I want to give you the opportunity to address the reviewer's main points of concern. If you are able to address those concerns and revise your manuscript accordingly, I am inclined to accept it. Reviewer 3's primary concerns were (1) that the manuscript lacked a systematic evaluation using accuracy and stability measures which are needed to judge the validity of the results and (2) that the manuscript lacked a comparison with any other existing related signatures like S1500 and L1000. Please focus on addressing these points in your revision.

Reviewer 1 ·

Basic reporting

1 Line 109-110, the author thinks the key genes and pathways can be used as a biomarker to control heavy metal pollutants. I think that the word “control heavy metal pollutant” is not accurate.

Experimental design

1 Authors obtained lots of genes to responsive to heavy metal through integrated bioinformatics analysis, such as 59 up-regulated genes and 109 down-regulated genes. But these theoretical results need more some experiments to verify. Therefore I suggest that authors should add some experimental verification to some critical gene expressions to responsible to heavy metals in their future work and try to screen the most sensitive genes as marker genes.

2 Line 118-120, the author has mentioned 5 kinds of heavy metal experiments, but I could not find the experiment details, so I think that the related experiment methods should be described in this paper.

Validity of the findings

no comment

Additional comments

The manuscript by Chao Niu et al. represents an interesting work regarding integrated bioinformatics analysis of As, Au, Cd, Pb and Cu heavy metal responsive marker genes through Arabidopsis thaliana GEO datasets. In this manuscript, the authors have obtained 168 differentially expressed genes (DEGs) which contained 59 up-regulated genes and 109 down-regulated genes through comparative bioinformatics analyses. The results are benefit to understand the potential molecular mechanisms or signaling pathways responsive to heavy metal stress in plants.

In my opinion, the manuscript can be accepted with minor revisions.

Reviewer 2 ·

Basic reporting

In the present manuscript submitted to PeerJ, Niu et al. analyze data from eleven microarrays to identify differentially expressed genes (DEGs) in Arabidopsis thaliana after different heavy metals exposures. So, the results and conclusions of this manuscript are of great interest because provides some clues for the detection of heavy metals pollutants in water and soil. This study is interesting, could be useful in the field and will be interesting for readers of PeerJ.
English is not my first language and I do not claim particular competence. However, I think that the paper is clearly written and logically structured. Overall, the references present are appropriate, well discuss and related with the results obtained in this work.

Experimental design

In my opinion, although my laboratory´s work does not focus on the field of microarrays, the experimental approach is adequate to analyze the DEGs that are common after exposure to several heavy metals. Until now, differentially expressed genes had only been analyzed individually after exposure to a single heavy metal.

Validity of the findings

In my opinion, the discussion seems to be well written. However, this section is weak and I think the authors should discuss, apart from the DEGs in common, the DEGs that are not common among all heavy metals. The discussion would benefit from a minor revision.

Additional comments

No comment

·

Basic reporting

Summary:
Authors in this study analyzed several microarray expression datasets towards developing a biomarker for detecting contaminations of a group of chemicals in Arabidopsis thaliana plant. The study applies differential gene expression analysis using mainly traditional approaches and summarizes finding of several genes. Regarding the approach, the work is based on a very typical bioinformatics analysis and no novelty is realized. The work lacks any systematic evaluation of the accuracy and stability. Also, there is no comparison with any existing method or previously developed notable signatures like S1500 and L1000.

Positives:
1- Applying bioinformatics for biomarker detection of environmental chemicals
2- Extending DEGs analysis with PPI networks and exploring more significant genes.

Negatives:
1- Approach is oversimplified for biomarker detection and analysis.
2- Lacking systematic evaluation using accuracy and stability measures needed to judge the validity of the results.
3- Lacking comparison with any other existing related signature such as S1500.

Experimental design

- The approach for finding differentially expressed genes is based on limma which is one of the standard statistical approaches used for ranking genes. While this is one of the well-known solutions, biomarker development generally goes beyond in an effort to tackle limiting assumption like linearity and independence of genes. Related and notable works of signature development are S1500 (toxic exposures related) and L1000 (drug treatment related). Both signatures and other works, propose more steps such as clustering gene space, dimensionality reduction, and expression space coverage. Authors are highly advised to:
--Study these developed solutions closely
--Compare with the signatures at the conceptual or performance level

- Another point regarding the approach, is the limited elaboration on common genes between the different subsets. DEGs were generated for each dataset but how much these groups share. This is very useful to understand stability and relevance (or importance) of the selected group of genes. Related to this point, is providing details on methods like RRA and possible variants of selecting a final group of genes.

Validity of the findings

- A strong limitation of this study is lacking any systematic evaluation of the accuracy and stability of results. Authors resorted mainly to enrichment analysis and related literature to highlight the significance of selected genes. However, established accuracy measures like the sensitivity of selected genes in defining toxic samples should be used. If samples are not labeled (i.e. phenotype is not known) to perform such evaluation, then, showing gene expression coverage using the selected genes will be useful (see L1000 evaluation). Other types of evaluations can be found in the literature as well. As an example, see the following study:
Wang, Pingping, et al. "A reduced transcriptome approach to assess environmental toxicants using
zebrafish embryo test." Environmental science & technology 52.2 (2018): 821-830.
For stability evaluation, were the 9 cores genes found using network construction step among the top 168 ranked earlier? PPI network was introduced as another way to choose genes and while this is true, it was not clear how results were related to the previous analysis.

- It was not clear from the result section, how specific the developed set of genes are for the Arabidopsis thaliana plant.

Additional comments

Minor points:
- For figure 3, what about a heatmap for all experiments together. For figure 4, it should be annotated with control vs. treatment (or exposure) and then, maybe the name of experiments.

- "the cluster heatmap of the top 200 DEGs identified was displayed by" --> How the top 200 were defined? Was p-value score average from different experiments or what?

- Not sure about a minor issue as in the compiled PDF file of the manuscript, two abstracts exist where the first one has some typos."detecting their expression in plants growing in those environments. anisms or signaling
... through detecting their expression in plants growing in those environments."!!

- "afterwards, it enters the animal" --> "Afterwards,"
"Then these datasets" --> "Then"," these datasets"
Review and fix other related typos.

- "However, these solutions are not convenient nor efficient" --> Elaborate briefly on this.

- In "Retrieval of gene expression profile data" sub-section please provide brief details
about the platforms processed and using which approach. As in the supplementary table 1,
RMA and MAS5.0 were applied to most of the datasets. How did you address combining
different types of normalizations??

- "in 20 raw were.." --> What "raw" refers to here?

As a side note, it was interesting to see that MAPK signaling pathway is among the top enriched ones. This complies with previous findings.

---

## Round 0.2 · Minor Revisions

Nice job of addressing reviewer comments and revising your manuscript.
There are a couple additional minor comments identified in the review of your revised manuscript. I hope you are willing to make these minor changes.

Reviewer 2 ·

Basic reporting

No comment

Experimental design

No comment

Validity of the findings

No comment

Additional comments

All the questions have been solved.

·

Basic reporting

Authors in this review have addressed most of the major comments.

Experimental design

It is appreciated that they have performed an experimental validation to support the previous findings. Experimental validation using qPCR is certainly a valuable inclusion and it supports the conclusion.

Validity of the findings

The findings were supported by several statistical analysis tools and experimental validation.

Additional comments

- For "mentioned above (Cd, Pb and Cu) or no mentioned above", change "mentioned above" and "no mentioned above" in all text to "training compounds" and "testing compounds" or "study compounds" and "external testing compounds" or other more clear expression. In addition, in Abstract and Experimental validation sections, list the chemicals "mentioned above (Cd, Pb and Cu) or no mentioned above (Zn and Cr)"

- In Introduction "..Library of Integrated Network-based Cellular Signatures (LINCS ) L1000 data", add citation to L1000:
Subramanian, Aravind, et al. "A next generation connectivity map: L1000 platform and the first 1,000,000 profiles." Cell 171.6 (2017): 1437-1452.
Please make sure to cite L1000 in text.

- Fix statement:
"The real time quantitative PCR (RT-qPCR) experiments on RNA collected "-->
"... RNA 'were' collected "

- Experimental validation: add "quantitative PCR (RT-qPCR) validation" to abstract.

---

## Round 0.3 · accepted · Accept

Thank you for your efforts in revising your manuscript to improve its readability and your willingness to do so.

#